# Differential Response to Sorafenib Administration for Advanced Hepatocellular Carcinoma

**DOI:** 10.3390/biomedicines10092277

**Published:** 2022-09-14

**Authors:** Song-Fong Huang, Sio-Wai Chong, Chun-Wei Huang, Heng-Yuan Hsu, Kuang-Tse Pan, Chien-Fu Hung, Tsung-Han Wu, Chao-Wei Lee, Chia-Hsun Hsieh, Ching-Ting Wang, Pei-Mei Chai, Ming-Chin Yu

**Affiliations:** 1Department of Surgery, New Taipei Municipal Tucheng Hospital, New Taipei City 23652, Taiwan; 2Department of Medical Imaging and Intervention, Linkou Chang Gung Memorial Hospital, Taoyuan City 333008, Taiwan; 3Department of Radiology, New Taipei Municipal Tucheng Hospital, New Taipei City 23652, Taiwan; 4Department of Surgery, Chang Gung Memorial Hospital, Linkou and Chang-Gung University, Taoyuan City 33305, Taiwan; 5Division of Hematology-Oncology, Department of Internal Medicine, New Taipei Municipal Tucheng Hospital, New Taipei City 23652, Taiwan; 6Department of Nuring, Chang Gung Memorial Hospital, College of Medicine, Chang Gung University, Taoyuan City 33302, Taiwan

**Keywords:** advanced HCC, sorafenib, target lesions, ALBI

## Abstract

Sorafenib has been used to treat advanced hepatocellular carcinoma (aHCC). However, there is no evidence for a response of different target lesions to sorafenib administration. Therefore, we aimed to evaluate the effect of sorafenib on various aHCC target lesions. The outcomes of sorafenib treatment on aHCC, i.e., treatment response for all Child A status patients receiving the drug, were analyzed. Of 377 aHCC patients, 73 (19.3%) had complete/partial response to sorafenib, while 134 (35.4%) and 171 (45.2) had a stable or progressive disease, respectively, in the first six months. Of the evaluated metastatic lesions, 149 (39.4%), 48 (12.7%), 123 (32.5%), 98 (25.9%), 83 (22.0%), and 45 (11.9%) were present in liver, bone, lung, portal/hepatic vein thrombus, lymph nodes, and peritoneum, respectively. The overall survival and duration of treatment were 16.9 ± 18.3 and 8.1 ± 10.5 months (with median times of 11.4 and 4.6, respectively). Our analysis showed poor outcomes in macroscopic venous thrombus and bone, higher AFP, and multiple target lesions. ALBI grade A had a better outcome. Sorafenib administration showed good treatment outcomes in selected situations. PD patients with thrombus or multiple metastases should be considered for sorafenib second-line treatment. The ALBI liver function test should be selected as a treatment criterion.

## 1. Introduction

Curative treatment for resectable hepatocellular carcinoma (HCC) leads to the best outcome, while recurrence, including intrahepatic, disseminated spread, or distant metastases, require systemic treatment, i.e., sorafenib and regorafenib target therapy [1,2,3,4,5]. The existence of several options for first- and second-line systemic treatments of advanced HCC improves the overall survival [6,7]. Regorafenib second-line treatment has been used for advanced HCC. This approach was shown to prolong overall survival to 32 months in a resource study [8,9]. However, there was no concensus about the survival benefit in different target lesions, such as bone, lung, lymph node and peritoneal metastasis [10,11,12]. Therefore, although HCC with different organ metastasis was assumed as stage IV, varying clinical responses should be studied.

Albumin-bilirubin (ALBI) grade has been shown to be an indicator of liver dysfunction [13,14]. A combination of ALBI and APRI showed superior predicting power for postoperative hepatic failure [15]. ALBI grade 2 was shown to be a significant negative predictor when patients were treated with eluting embolic chemoembolization [16]. Ramucirumab has been shown to be of survival benefit in ALBI grade 1, and patients with grade 2 or 3 expressed liver-specific adverse events in a REACH study [17]. ALBI grade was found to be a predictor for HCC outcome in regional ablation therapy and systemic administration [18]. Alpha-fetoprotein (AFP) is a standard diagnostic marker; serum level over 400 ng/mL is a selection criterion for sequential therapy with ramucirumab [19]. The role of AFP in tyrosine kinase inhibition is debated [20]. Although several biomarkers have been developed for clinical application, AFP remains the most important one [21,22].

This study is the first to explore the organs which are susceptible to target lesions. It examined the differences of responses and long-term outcomes of sorafenib systemic treatment in advanced HCC. Liver fucntion, ALBI grading and AFP response and clinical relevance were also studied.

## 2. Materials and Methods

### 2.1. Patients and Samples

This study was approved by the Institute Review Board (IRB) of Chang Gung Memorial Hospital (CGMH), Linkou (IRB 201600513B0). Since 2012, when the National Health Insurance system approved the administration of sorafenib as a first-line treatment for advanced HCC, 377 HCC patients have been treated. Treatment indications included extrahepatic spreading (EHS), macroscopic venous invasion (MVI), or refractory response to transarterial chemoembolization (TACE). All enrolled patients showed good performance status, with ECOG scores between 0 to 1 and Child-Pugh A status, and received sorafenib-based drugs as target therapy with or without combination treatment. The clinical and pathological variables were collected for analysis. Progression-free and overall survival were compared using the log-rank test and Kaplan-Meier survival analysis. The clinical response to sorafenib was measured according to the RECIST criteria and classified as complete response (CR), partial response (PR), stable disease (SD), or progressive disease (PD) [23].

### 2.2. Statistical Analysis

Categorical data were analyzed using the chi-square test or Fisher’s exact test. Continuous variables were analyzed using a t-test. Survival rates in each group were determined by the Kaplan-Meier method. Differences between groups were analyzed using the log-rank test. All calculated *p*-values were two-tailed, with the significance defined at the 95% level (*p* < 0.05). Statistical analyses were performed using SPSS statistical software version 19.0 (SPSS, Inc., Chicago, IL, USA).

## 3. Results

### 3.1. Analysis of Clinical Response to Sorafenib Administration

In the CR, PR, SD, and PD clinical responses of the 377 patients to sorafenib, the outcomes were 18 (4.8%), 55 (14.6%), 133 (35.3%), and 171 (45.4%), and the overall clinical responses were 15 (4.0%), 12 (3.2%), 62 (16.4%) and 288 (76.4%), respectively. The objective response rate in the first 6 months was 54.6%. Positivity of hepatitis B and C infection was 61.3% and 29.7%, respectively, all in Child-Pugh A status. Within the sample, 133 (35.5%) showed AFP levels more than 400 ng/mL, and 215 (57.4%), 150 (40.1%), and 9 (2.4%) showed ALBI grades 1, 2, and 3, respectively. The most common extrahepatic spread was to bone, lung, peritoneum, and lymph nodes, in the form of target lesions; 157 (41.6%) patients had multiple target lesions (Table 1). The mean duration of treatment and overall survival were 8.1 ± 10.5 and 16.9 ± 18.3 months, respectively. The overall survival rate was 84.8%, 68.5%, 51.6%, 39.5%, 38.5%, and 33.5% in 6 months and 1, 2, 3, 4 and 5 years, respectively. The progression-free survival rate was 48.4%, 27.9%, 13.3%, 11.1%, 7.1%, and 5.4% in 6 months and 1, 2, 3, 4, and 5 years, respectively (Figure 1). There were significant differences in aspartate transaminase (AST), AFP and ALBI grade among the best responders (*p* = 0.002, 0.021, and 0.011, respectively).

Based on the guidelines of the Taiwan Liver Cancer Association and National Health Insurance of Taiwan, the indications for sorafenib as a first line prescription are extrahepatic spreading (EHS) and MVI involved the first generation and TACE refractoriness (64.7%, 21.0%, and 14.3%, respectively). There were no significant differences in indication, dosage or sequential treatment among the best clinical responders. Fourteen cases underwent surgery or radiofrequency ablation (RFA), while 81 received TACE, 72 received radiotherapy (RT) and 11 received chemotherapy as part of a combination treatment. The combination of sorafenib with surgical resection, TACE, RFA, or radiotherapy could significantly increase the probability of favorable oncologic outcome. There were significantly poorer responses in cases with bone metastasis, lung metastasis and/or multiple target lesions.

### 3.2. Differential Oncological Outcomes Based on the Presence of Organ-Specific Target Lesions and ALBI Grade

Our analysis of progression-free survival (PFS) showed that significantly poorer prognostic factors included multiple targets, bone, lung, MVI, ALBI grade 2 or 3, AFP > 400 ng/mL, AST two times elevation and combined treatment (Figure 2; *p* = 0.009, 0.021, 0.036, 0.038, <0.001, 0.002, 0.004, and 0.019, respectively). The Cox regression multivariate analysis showed that bone, lung, MVI, ALBI grade 2 or 3, AFP > 400 ng/mL and combined treatment were independent prognostic factors (Table 2, *p* = 0.001, 0.008, 0.008, <0.001, 0.008 and 0.026, respectively).

In a subgroup analysis, when patients with multiple targets were excluded from the single-target subgroup analysis, there were significantly better outcomes for patients with LN metastasis. Patients with MVI still demonstrated the worst outcome in both PFS and OS. Furthermore, those with bone or lung metastases showed significant survival benefits in OS but not PFS; this was attributed to sequential therapy (see Figure 3).

### 3.3. Sequential Target Therapy Resulted in Better Survival Outcome

Of the 345 patients with survival follow-up when PD was recorded, 132 had the option of to taking the second line treatment, including target therapy, chemotherapy, ablation, surgery and checkpoint inhibitors. The survival benefit was identified in OS for sequential therapy. Different survival responses were observed with different modality treatment bases. A better survival outcome was noted in sequential target therapy and ablation/surgical treatments (Figure 4).

## 4. Discussion

Over the past decade, new systemic treatments for advanced HCC have led to improved patient outcomes. Clinical practice has several treatment regimens for first- and second-line therapies [24]. A combination of immune checkpoint inhibitors and tyrosine kinase inhibitors could be a breakthrough treatment modality, but defects in interferon-γ or insufficient tumor antigen immunosuppressive cells in the tumor microenvironment develop resistance to immune checkpoint inhibitors [25,26]. Sorafenib and other TKIs have been used globally, and molecular biomarkers and genome changes in pathobiological issues have been emphasized [27]. However, varying outcomes in different metastatic locations have been studied. The current evidence indicates the worst outcomes with bone and lung metastases. Patients with aHCC usually had multiple target lesions for treatment (41.6%), and it is hard to determine the cause of cancer-related mortality and organ involvement. However, the LN and peritoneal appear to benefit or show non-inferiority with sorafenib administration, compared with cases in which other organs are involved.

AFP is not only a diagnostic serologic surveillance lab test, but also a tool for the prognostic analysis of surgical outcomes [23]. AFP > 400 ng/mL and AFP response in the REACH study showed better outcomes in OS and PFS, demonstrating the importance of biomarker studies for HCC [28]. Elevated AFP represents a subgroup in HCC tumor heterogeneity [29]. A combination of ALBI grade and AFP level was used as a tool for patient outcome prediction in 88 real-world retrospective cohort study cases [21]. All treated patients showed Child-Pugh grade A status, and ALBI grading was the best evaluation factor for liver function tests. Patients with two-fold elevations of AST were not enrolled in the early clinical phase III studies, and few reports have focused on the impact of chronic hepatitis. However, an elevation of AST representing hepatitis B reactivation and antiviral treatment could prevent prevent reactivation and prolong overall survival for hepatic artery infusion chemotherapy in HCC [30,31,32]. Therefore, AST elevation was also a negative predicting factor, suggesting that patients with liver damage should be carefully monitored in real-world practice.

Combination and multimodality treatments offer a better treatment option for advanced HCC. Radiation or ablation therapies are commonly used for local tumor control. Treating patients at the optimum physiologic status with minimal damage could offer better oncologic outcomes. Combination immunotherapy with immune checkpoint inhibitors (ICI) yielded limited data, and no definite conclusion could be reached. However, development in ICI and tyrosine kinase inhibitors is expected to provide new primary treatment choices in the future.

Though all metastatic HCC could be treated with TKI or dual therapy, no evidence of treatment efficiency at different metastatic sites was found. TMmain venous tumor thrombosis showed poor outcome. Hence, a combined treatment with ICI could be considered. Surgical resection could be considered a combined or sequential strategy for single metastatic lung lesions to improve outcomes. Though HCC with LN metastasis showed no significant benefit in total, better outcomes were observed in single target lesion analysis in our study. A longer DOR was reported in [33,34].

Our study has some limitations. Firstly, it is not a prospective study. Further, there was some bias in the study period. Treatment options have increased in recent years, with other TKIs or dual therapies becoming available as first-line treatments. However, the study revealed that patients with different metastatic target lesions could present different outcomes, providing justification for alternative treatment protocols.

## 5. Conclusions

Advanced HCC with sorafenib as a first-line regimen had poor outcomes in bone or lung metastasis, MVI, ALBI grade 2 or 3, and AFP > 400 ng/mL. Combined treatment of a local tumor control showed better outcomes. Add-on sorafenib treatment in lung or bone metastasis should be considered.

Although the number of cases enrolled in this study was limited, the data offer some insights. Firstly, independent poor prognostic factors included bone, lung, MVI, ALBI grade 2 or 3, and AFP > 400 ng/mL; combined treatment offered better outcomes. Secondly, bone and lung involvement indicated a higher probability of progression, but sequential therapy including ablation/surgery and second-line targeting resulted in relatively higher long-term survival outcomes.

## Figures and Tables

**Figure 1 biomedicines-10-02277-f001:**
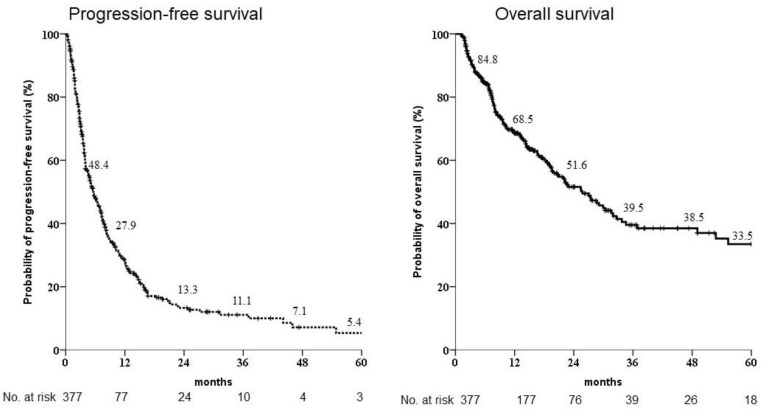
Kaplan-Meier estimates of survival rates of the 377 patients enrolled for sorafenib administration as first-line treatment. The progression-free survival rates at 1, 2 and 3 years were 27.9%, 13.3% and 11.1%, respectively, whereas the overall survival rates after 1, 2 and 3 years were 68.5%, 51.6% and 39.5%, respectively.

**Figure 2 biomedicines-10-02277-f002:**
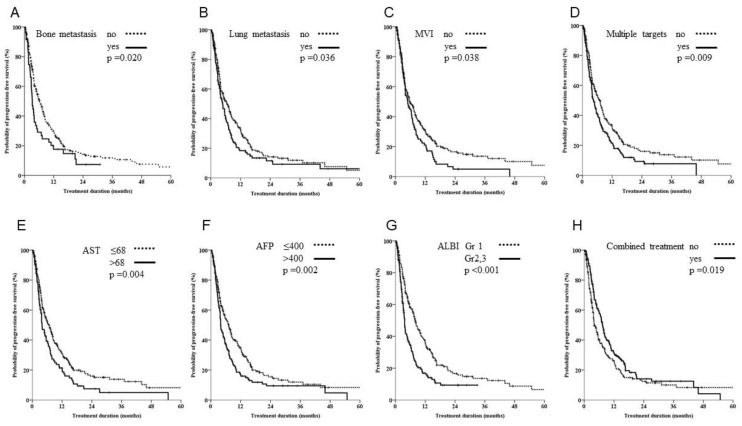
Kaplan-Meier estimates of survival rates stratified by different organ metastasis, vascular involvement, hepatitis or combined treatment. (**A**–**D**) Patients with bone metastasis, lung metastasis, macroscopic venous invasion (MVI) or multiple lesions had significantly poorer outcomes in terms of progression-free survival. (**E**,**F**) Patients with aspartate transaminase (AST) levels twice those of normal values and alpha fetoprotein (AFP) levels of >400 ng/mL had significantly poorer outcomes in terms of progression-free survival. (**G**) Patients with grade III albumin-bilirubin had poor outcomes compared with patients with grade I/II. (**H**) A combination of sorafenib and ablation treatment yielded significantly better outcomes.

**Figure 3 biomedicines-10-02277-f003:**
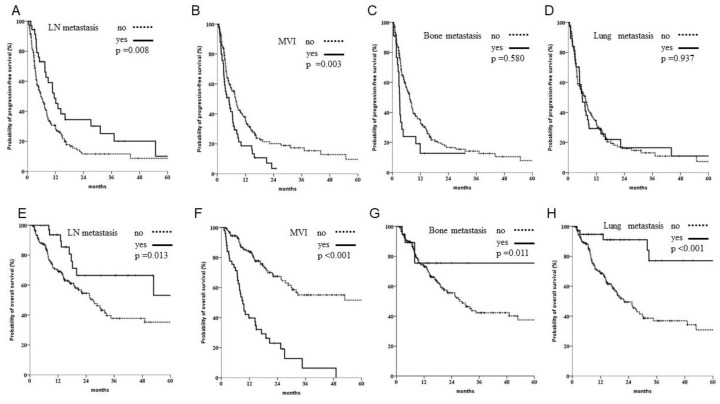
Subgroup analysis of progression and overall survival when patients with multiple targets were excluded. (**A**–**D**) Of 220 patients with single organ involvement, those with lymph node (LN) metastasis had better outcomes, whereas those with bone metastasis or macroscopic venous invasion (MVI) had worse outcomes. (**E**–**H**) Significant better OS was noted in patients with single organ involvement, including LNs, bones and lungs, but worse outcomes were observed in patients with MVI.

**Figure 4 biomedicines-10-02277-f004:**
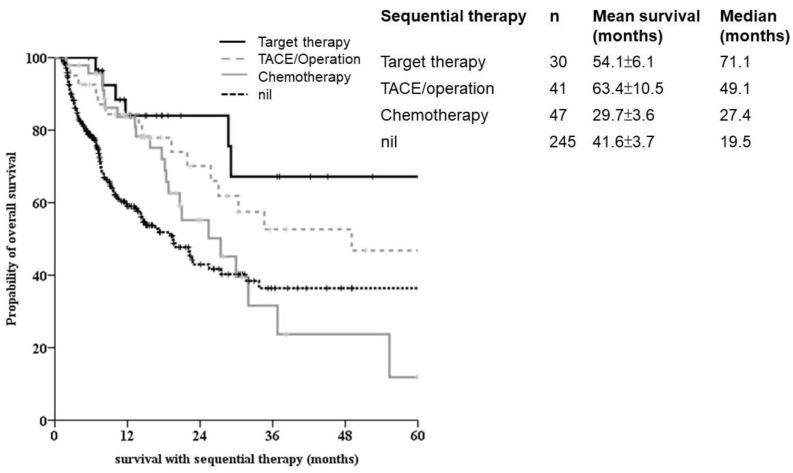
Sequential therapy for advanced HCC. Kaplan-Meier survival analysis showed significantly better outcomes in the tyrosine kinase inhibitor group (solid line). Patients with chemotherapy-based second line-treatment for aHCC showed significantly poorer outcomes.

**Table 1 biomedicines-10-02277-t001:** Demographic data of advanced HCC patients receiving sorafenib, and correlation with clinical responses.

Variables	All	Best Clinical Response (CR PR)	Best Clinical Response (SD)	Best Clinical Response (PD)	*p*-Value
	*n* = 377	*n* = 73 (19.3)	*n* = 133 (35.3)	*n* = 171 (45.4)	
Age (years)	62.1 ± 12.1	60.4 ± 13.3	64.4 ± 10.9	62.2 ± 12.3	0.066
Gender (male)	311 (82.5)	63 (86.3)	107 (80.5)	141 (82.5)	0.572
Comorbidity(yes)	170 (45.1)	28 (38.4)	65 (48.9)	77 (45.0)	0.349
HBV positive	231 (61.3)	42 (57.5)	79 (59.4)	110 (64.3)	0.522
HCV positive	112 (29.7)	27 (37.0)	41 (30.8)	44 (25.7)	0.199
WBC (1000/μL)	6040.8 ± 2553.1	5918.1 ± 3550.2	5927.1 ± 2153.5	6163.6 ± 2333.3	0.669
AST (U/L)	59.9 ± 44.2	49.5 ± 30.0	54.8 ± 38.5	68.5 ± 51.7	0.002 *
ALB(g/dL)	4.0 ± 0.5	4.0 ± 0.6	4.1 ± 0.5	3.9 ± 0.5	0.002 *
Bilirubin (mg/dL)	0.9 ± 0.7	0.9 ± 0.5	0.8 ± 0.4	0.9 ± 0.9	0.296
Platelet (103/μL)	164.8 ± 82.3	153.6 ± 72.1	159.9 ± 71.5	173.3 ± 93.1	0.161
ALBI grade 1	215 (57.5)	36 (63.0)	87 (66.4)	82 (48.2)	0.011 *
2	150 (40.1)	26 (35.6)	43 (32.8)	81 (47.6)	
3	9 (2.4)	1 (1.4)	1 (0.8)	7 (4.1)	
AFP (ng/mL)	14,094.3 ± 65,809.4	8165.9 ± 28,501.9	4154.2 ± 37,767.2	24,239.9 ± 88,872.8	0.021 *
AFP (>400 ng/mL)	133 (35.5)	21 (28.8)	32 (24.4)	80 (46.8)	<0.001 ***
Cirrhosis	177 (56.4)	30 (56.6)	56 (50.0)	91 (61.1)	0.203
Target lesions					
Liver	148 (39.3)	25 (34.2)	56 (42.1)	67 (39.2)	0.543
Bone	48 (12.7)	7 (9.6)	11 (8.3)	30 (17.5)	0.037 *
Lung	123 (32.6)	14 (19.2)	43 (32.3)	66 (38.6)	0.012 *
Thrombus	98 (26.0	16 (21.9)	31 (23.3)	51 (29.8)	0.296
LN	83 (22.0)	18 (24.7)	31 (23.3)	34 (19.9)	0.644
Peritoneum	45 (11.9)	12 (16.4)	14 (10.5)	19 (11.1)	0.413
Brain	4 (1.1)	2 (2.7)	0	2 (1.2)	NA
Multiple	157 (41.6)	22 (30.1)	52 (39.1)	83 (48.6)	0.022 *
AE	90 (23.9)	16 (21.9)	35 (26.3)	39 (22.8)	0.706

HBV: hepatitis B virus; HCV: hepatitis C virus; AST: aspartate aminotransferase; ALB: albumin; AFP: alpha-fetoprotein; AE: adverse events. * *p* < 0.05, *** *p* < 0.001. Percentages are given in the brackets.

**Table 2 biomedicines-10-02277-t002:** Results of univariate and multivariate Cox regression analysis for progression-free survival following sorafenib treatment for 377 advanced HCC patients.

Variable	Univariate Analysis			Multivariate Analysis		
	HR	95% CI	*p*-Value	HR	95% CI	*p*-Value
Age (years), >65 vs. ≤65	0.853	0.674–1.078	0.183			
Sex (M/F), M vs. F	0.735	0.775–1.430	0.741			
Comorbidity, Yes vs. No	1.090	0.864–1.375	0.469			
HBV, Yes vs. No	1.194	0.939–1.518	0.148			
HCV, Yes vs. No	0.833	0.646–1.074	0.159			
Multiple targets, Yes vs. No	1.365	1.079–1.727	0.009 **	0.941	0.717–1.236	0.663
Liver (no MVI), Yes vs. No	1.049	0.824–1.334	0.699			
Bone, Yes vs. No	1.478	1.060–2.059	0.021 *	1.868	1.287–2.712	0.001 **
Lung, Yes vs. No	1.298	1.017–1.657	0.036 *	1.509	1.111–2.041	0.008 **
Lymph Nodes, Yes vs. No	0.791	0.594–1.053	0.108			
Peritoneum, Yes vs. No	0.875	0.608–1.258	0.471			
MVI, Yes vs. No	1.314	1.015–1.700	0.038 *	1.511	1.115–2.048	0.008 **
Cirrhosis, Yes vs. No	1.170	0.903–2.558	1.513			
ALBI grades, 2, 3 vs. 1	1.694	1.369–2.094	<0.001 ***	1.674	1.292–2.169	<0.001 ***
AFP (400 ng/mL), >400 vs. ≤400	1.446	1.140–1.834	0.002 **	1.408	1.094–1.811	0.008 **
AST (IU/L) 2 ULN, >68 vs. ≤68	1.452	1.126–1.873	0.004 **	1.237	0.932–1.642	0.140
ALT (IU/L) 2 ULN, >72 vs. ≤72	1.238	0.932–1.644	0.140			
Combined treatment, Yes vs. No	0.755	0.596–0.955	0.019 *	0.754	0.587–0.967	0.026 *

HR, hazard ratio; 95% CI, 95% confidence interval of hazard ratio. Disease free survival was calculated by univariate and multivariate Cox regression analysis. * *p* < 0.05, ** *p* < 0.01, *** *p* < 0.001.

## Data Availability

Not applicable.

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
