# Peer review of "Differential Response to Sorafenib Administration for Advanced Hepatocellular Carcinoma"

_biomedicines, 2022, doi:10.3390/biomedicines10092277_

Round 1

Reviewer 1 Report

Hepatocellular carcinoma (HCC) is a prevalent malignancy worldwide and one of the difficult cancers to cure completely. Several kinase inhibitors and immune checkpoint inhibitor have been already applied for unresectable/advanced HCC, it is however hard to say that their clinical effects are still satisfied. Sorafenib was the first molecular-targeting drugs for HCC and used worldwide broadly even now. The authors would like to have also demonstrated the differential effectiveness of sorafenib using the rich number of HCC patients in the authors’ hospital. The reviewer is also interested in these results about the relationship between HCC and sorafenib, there are however some questions to clear up more as below: 

#1: What is the novelty of this report? While there has been already a number of reports about the treatment effects of sorafenib for HCC, what are the strongest point that the authors would like to claim in this study and the specific differences from the former reports?

#2: What is the null hypothesis for the correlation analysis in table 1?

#3: In the “Introduction” section, the reviewer could not clearly understand the clinical question to solve for the authors in this study. Please clarify the background and the aim of this study. 

#4: It was hard for the reviewer to believe and understand that ALBI (albumin-bilirubin) grade (G) and aspartate transaminase (AST, E) were stronger prognostic factors than distant metastasis such as bone (A) and lung (B) in figure 2.

#5: The reviewer thought that there were some inconsistency between figure 3 and table 2 because the independent prognostic factors were different. For example, there was statistical differences in LN metastasis (Figure 3A), lymph nodes metastasis was not however a strong prognostic factor in table 2.      

#6: Please define “target therapy” clearly in the “Materials and Methods” section.

Reviewer 2 Report

The article aims to evaluate the effect of sorafenib on various aHCC target lesions. Nevertheless, lot of information is lacking and an effort to complete it should be attempted.
Methods: 
- Please provide more information about the combained treatments.Curative treatments or palliative treatments performed for local tumor control

- HCC features, liver parenchima involment,extrahepatic features of metastatic disease should be described and valuated. 

Round 2

Reviewer 1 Report

The reviewer thought that the authors sincerely revised the paper according to the reviewer’s comments and questions. The reviewer has no other comments and questions for present revised version of this article.